# The Hidden Dangers of Sedentary Living: Insights into Molecular, Cellular, and Systemic Mechanisms

**DOI:** 10.3390/ijms251910757

**Published:** 2024-10-06

**Authors:** Daniel Guerreiro Diniz, João Bento-Torres, Victor Oliveira da Costa, Josilayne Patricia Ramos Carvalho, Alessandra Mendonça Tomás, Thaís Cristina Galdino de Oliveira, Fernanda Cabral Soares, Liliane Dias e Dias de Macedo, Naina Yuki Vieira Jardim, Natáli Valim Oliver Bento-Torres, Daniel Clive Anthony, Dora Brites, Cristovam Wanderley Picanço Diniz

**Affiliations:** 1Laboratório de Microscopia Eletrônica, Instituto Evandro Chagas, Seção de Hepatologia, Belém 66.093-020, Pará, Brazil; danielguerreirodiniz@gmail.com; 2Núcleo de Pesquisas em Oncologia, Hospital Universitário João de Barros Barreto, Universidade Federal do Pará, Belém 66.073-005, Pará, Brazil; cwpdiniz@gmail.com; 3Laboratório de Investigações em Neurodegeneração e Infecção, Hospital Universitário João de Barros Barreto, Universidade Federal do Pará, Belém 66.073-005, Pará, Brazil; bentotorres@gmail.com (J.B.-T.); violiveiradacosta@gmail.com (V.O.d.C.); josilayne.28.patricia@gmail.com (J.P.R.C.); alessandratomas@ufpa.br (A.M.T.); thais_cgo@hotmail.com (T.C.G.d.O.); fefacsoares@hotmail.com (F.C.S.); anediasdias@hotmail.com (L.D.e.D.d.M.); naina.jardim@gmail.com (N.Y.V.J.); 4Programa de Pós-Graduação em Ciências do Movimento Humano, Universidade Federal do Pará, Belém 66.050-160, Pará, Brazil; 5Campus Samabaia, Universidade Federal de Goiás (EBTT), CEPAE, Goiânia 74.001-970, Goiás, Brazil; 6Faculdade de Ceilândia, Ceilândia, Universidade de Brasília, Brasília 72.220-900, Brazil; 7Campus Tucurui, Universidade do Estado do Pará, Tucurui 68.455-210, Pará, Brazil; 8Programa de Pós-Graduação em Neurociências e Biologia Celular, Instituto de Ciências Biológicas, Universidade Federal do Pará, Belém 66.075-110, Pará, Brazil; 9Laboratory of Experimental Neuropathology, Department of Pharmacology, University of Oxford, Oxford OX1 2JD, UK; daniel.anthony@pharm.ox.ac.uk; 10Faculty of Pharmacy, Department of Pharmaceutical Sciences and Medicines, Universidade de Lisboa, 1649-003 Lisbon, Portugal; dbrites@ff.ulisboa.pt; 11Faculty of Pharmacy, Research Institute for Medicines (iMed.ULisboa), Universidade de Lisboa, 1649-003 Lisbon, Portugal

**Keywords:** age-related cognitive decline, immunosenescence, neurodegeneration, neuroinflammation, oxidative stress, physical exercise, non-pharmacological interventions, therapeutic interventions, synaptic dysfunction

## Abstract

With the aging of the global population, neurodegenerative diseases are emerging as a major public health issue. The adoption of a less sedentary lifestyle has been shown to have a beneficial effect on cognitive decline, but the molecular mechanisms responsible are less clear. Here we provide a detailed analysis of the complex molecular, cellular, and systemic mechanisms underlying age-related cognitive decline and how lifestyle choices influence these processes. A review of the evidence from animal models, human studies, and postmortem analyses emphasizes the importance of integrating physical exercise with cognitive, multisensory, and motor stimulation as part of a multifaceted approach to mitigating cognitive decline. We highlight the potential of these non-pharmacological interventions to address key aging hallmarks, such as genomic instability, telomere attrition, and neuroinflammation, and underscore the need for comprehensive and personalized strategies to promote cognitive resilience and healthy aging.

## 1. Introduction

Cognitive decline associated with aging, a widespread and critical public health concern, is worsened by today’s sedentary lifestyle. Extended periods of sitting or inactivity are now acknowledged as key factors contributing to cognitive decline in older individuals. Studies have shown that excessive sedentary behavior is associated with structural brain changes, particularly in areas critical for memory and cognitive function, such as the hippocampus [1,2]. Sedentary behavior is defined as any waking activity that involves minimal energy expenditure, such as sitting or lying down for extended durations and has been strongly linked to negative health outcomes, including cognitive decline [3]. In contrast, an active lifestyle involves regular physical activities, including routine movements and planned exercises, which promote overall health and brain function. Physical exercise, a subset of physical activity, involves structured, repetitive movements designed to improve or maintain fitness, with recommendations suggesting 150–300 min of moderate to vigorous exercise per week to reduce cognitive decline risks (World Health Organization, accessed on 28 September 2024) (https://www.who.int/publications/i/item/9789240015128).

Despite widespread recognition of the benefits of physical activity, many individuals continue to lead inactive lives, influenced by societal structures that promote inactivity and dissociate physical activity from food intake [4]. Europeans, for instance, spend 40% of their leisure time watching television, while Americans spend 55% of their leisure time in sedentary activities (sitting and TV viewing), averaging 7.7 h per day [5]. This sedentary behavior and poor nutrition significantly contribute to cognitive decline in the aging process [6,7,8,9].

As life expectancy rises, age-related cognitive decline becomes increasingly significant [10]. It affects a large portion of the elderly population, with incidence rates 70% higher than dementia alone [11]. The aging process affects cognitive function to varying degrees, influencing domains such as memory, attention, and executive function [12,13]. Individuals exhibit distinct aging trajectories, shaped by their unique genotypes—encompassing metabolic, immune, hepatic, and nephrotic systems—along with other factors, including lifestyle and environmental exposures [14,15,16,17]. Lifestyle changes are known to modify an individual’s aging trajectory and can interact with one or more of the genotypes that are known to influence the aging process [18,19,20,21].

While much attention has been focused on physical activity, it is essential to distinguish between physical activity and exercise. Physical activity refers to any movement of the body resulting in energy expenditure and includes domestic, occupational, transportation, and leisure activities [22]. Exercise is a specific form of physical activity intentionally planned, structured, and repetitive to enhance or preserve physical fitness [23]. This distinction is critical, as evidence suggests that habitual physical activity and structured exercise may impact cognitive decline differently. Studies show that while general physical activity, such as walking or gardening, provides cognitive benefits, structured exercise programs, particularly those that include aerobic and resistance training, demonstrate more significant improvements in cognitive function in older adults [24,25,26].

It is essential to clarify that sedentary behavior is not simply the absence of physical activity. Instead, it is a distinct behavioral pattern characterized by prolonged periods of sitting or inactivity, which poses unique health risks, even for individuals who meet recommended physical activity guidelines. Recent research has emphasized that even individuals who meet the recommended physical activity levels may experience cognitive impairments if they spend excessive time in sedentary behaviors [27]. This highlights a significant limitation in current research, as many studies primarily focus on physical activity levels without adequately addressing the detrimental effects of prolonged inactivity [28]. Thus, future research must consider increasing physical activity and reducing sedentary time to improve cognitive outcomes in aging populations [29]. Conversely, an over-active lifestyle, characterized by excessive physical exertion without adequate rest, can also lead to physical and cognitive strain [30]. These lifestyle behaviors, including sedentary, active, and over-active lifestyles, must be carefully balanced, particularly in older adults, to support healthy cognitive aging.

Healthy lifestyle behaviors encompass physical activity, nutrition, mental well-being, and social engagement, all of which contribute to cognitive health in aging populations [31]. These behaviors must be understood in the context of aging trajectories, where different combinations of physical and cognitive activities and balanced nutrition can mitigate age-related cognitive decline.

The advantages of physical activity and exercise go beyond merely preventing cognitive decline. Even without structured exercise, physical activity has been linked to improved neurogenesis, synaptic plasticity, and reduced neuroinflammation, all contributing to overall cognitive health [26,32]. In contrast, structured exercise interventions have been associated with more significant increases in brain volume and function, particularly in regions such as the hippocampus, which is critical for memory and learning [33]. This distinction highlights the need to consider the quantity and type of physical activity when evaluating strategies to mitigate cognitive decline [32].

Cognitive and functional decline ranges from mild cognitive impairment (MCI) to severe conditions such as Alzheimer’s disease [34]. MCI is a transitional phase where cognitive difficulties are evident but not severe enough to disrupt daily functioning substantially [35,36,37,38]. However, MCI can progress to Alzheimer’s disease, characterized by substantial memory loss, impaired reasoning, and behavioral changes [39,40]. Understanding the progression from normal aging to MCI and eventually to Alzheimer’s disease is crucial for early diagnosis and intervention. Lately, mild behavioral impairment, an emergent and persistent neuropsychiatric symptom in individuals at risk for cognitive decline, was found to be prevalent in subjects with MCI and Alzheimer’s disease [41].

Longitudinal studies have shown that cognitive decline is often preceded by subtle changes in cognitive performance and brain structure, emphasizing the importance of early detection and monitoring [42]. These studies offer valuable insights into the risk factors and progression of cognitive decline, emphasizing the interaction between genetics, environmental factors, and lifestyle choices [43]. For more effective risk reduction, it is essential to consider individual lifestyle factors and the broader social-ecological public health perspective [44]. However, there remain gaps in the current literature, particularly in understanding how sedentary behavior interacts with other lifestyle factors like diet and social engagement in cognitive aging. More research is needed to address these limitations and develop comprehensive intervention strategies that target both physical activity and sedentary behavior, recognizing the multifaceted nature of the issue [45].

This integrative review aims to dissect the complex interplay of molecular, cellular, and systemic mechanisms contributing to age-related cognitive decline. Additionally, it highlights the importance of reducing a sedentary lifestyle by examining the effects of non-pharmacological interventions such as cognitive, multisensory, and motor stimulation. By synthesizing empirical evidence from experimental models and human studies, this review seeks to identify the essential molecular signatures that explain the therapeutic effects of these stimulation programs in reducing the progression of age-related cognitive decline.

## 2. Cognitive Decline Risk Factors and Aging Models

Research over the past decade has advanced our understanding of the risk factors associated with age-related cognitive decline [42,46,47,48,49,50,51,52]. Epidemiological studies have identified several modifiable and non-modifiable risk factors. Non-modifiable risk factors include age, genetics, and a family history of dementia. Modifiable risk factors involve lifestyle choices and health conditions, including physical activity, diet, cardiovascular health, education, and social interactions [42,53]. Among these, physical activity is one of the most significant contributors to cognitive health. Regular physical exercise has been linked to a lower risk of cognitive decline, mainly due to its positive effects on cardiovascular health and its role in enhancing neurogenesis and synaptic plasticity [30,54,55,56,57,58,59]. Physical activity is also thought to mitigate neuroinflammation, which is linked to cognitive impairment in aging [60,61,62,63].

Physical activity is essential for preserving cognitive health, with research emphasizing its protective effect by boosting cognitive reserve. More physically active individuals demonstrate greater resistance to cognitive decline, likely due to improved neurovascular health and increased brain plasticity [30]. This reserve allows individuals to maintain cognitive function despite age-related changes in the brain. The positive effects of physical activity go beyond improving cardiovascular health; they also directly influence the structural integrity of brain areas associated with memory and executive function, such as the hippocampus and prefrontal cortex [45]. This evidence highlights the importance of regular physical activity in preventive strategies for dementia.

Recent advancements in understanding the biological processes contributing to cognitive decline have provided invaluable insights into the cellular and molecular alterations of aging [64]. Experimental models, particularly in rodents, have facilitated controlled manipulation of genetic, environmental, and pharmacological factors, identifying critical molecular pathways involved in cognitive decline [65,66]. Rodent studies have elucidated mechanisms such as synaptic plasticity, neurogenesis, and the impact of stress and diet on cognitive function [67]. Interestingly, it has been demonstrated that whole-body clearance of senescent cells can alleviate age-related increases in basal ‘brain inflammation’ and cognitive impairment in murine models [68], which highlights the role of the host response to the degenerative process as a contributor to the loss of cognitive function with age.

Human-induced pluripotent stem cells and transdifferentiated cells from aged donors and patients as aging models in vitro may also help to unravel the crosstalk between aged and proliferative cells to understand how aging and disease develop [69] and how the responses can be reprogrammed in a dish [70]. Additional human studies are required to provide the necessary validation and context to the experimental studies. Research using neuroimaging, genomics, and postmortem brain analyses has uncovered patterns of brain aging and cognitive decline, facilitating the application of insights gained from animal models to human conditions [47,71]. Magnetic resonance imaging (MRI) and positron emission tomography (PET) have been instrumental in visualizing both structural and functional changes in the aging brain [38,72,73]. Medical imaging, particularly modalities like MRI and PET scans, has become increasingly valuable in identifying structural and functional changes in the brain associated with cognitive decline. Evidence suggests that imaging can reveal the cumulative effects of modifiable risk factors such as hypertension, diabetes, and smoking on brain aging. For example, Cumplido-Mayoral (2024) investigated the mediating effect of the difference between chronological age and MRI-derived brain age on the relationship between risk factors for Alzheimer’s disease and cognitive decline in individuals aged 45 and 65 who were cognitively unimpaired [74]. This study suggests that modifiable risk factors significantly impact age-related cognitive decline, reinforcing the importance of early intervention strategies to slow these changes [75].

Additionally, as Lin et al. (2023) demonstrate [76], integrating imaging biomarkers with cognitive performance data—such as those from the UK Biobank—provides a comprehensive framework for understanding sustainable cognitive aging. Imaging markers such as cortical thickness and white matter integrity are directly affected by modifiable risk factors, which can accelerate brain aging and heighten the risk of cognitive decline. By quantifying these imaging markers, researchers can monitor brain health over time, making it a valuable tool for tracking brain maintenance and potentially mitigating the impact of these risk factors on cognition. These studies have identified brain regions particularly vulnerable to aging and have correlated these changes with cognitive performance. Advances in PET imaging have allowed the in vivo visualization of amyloid and tau pathology, directly linking molecular changes to cognitive decline [47,77]. ^18^Fluoro-deoxy-glucose (FDG)-PET imaging has also been used to evaluate the alterations in cerebral glucose metabolism in MCI and was found to differentiate progressive MCI from stable MCI [78].

Genomic studies have identified genetic variants associated with an increased risk of cognitive decline and Alzheimer’s disease. Genome-wide association studies (GWAS) have uncovered numerous risk loci, including the well-known APOE ε4 allele, which is known to elevate the risk of Alzheimer’s disease significantly [79]. These genetic findings have provided some insights into the molecular pathways involved in cognitive decline, highlighting the critical roles of cholesterol metabolism, immune response, and synaptic function [42] though the precise role of the ε4 allele has remained an enigma that needs to be solved. Other research that has examined telomere length has underscored its potential role in cognitive health, suggesting that the preservation of telomere length has cognitive benefits among aged individuals at risk of dementia [80].

Postmortem brain analyses have confirmed and extended findings from neuroimaging and genomics studies, revealing the accumulation of amyloid plaques, neurofibrillary tangles, and other pathological features associated with Alzheimer’s disease [71,81,82]. These studies have also identified changes in neurotransmitter systems, synaptic density, and neuroinflammatory markers, providing a comprehensive picture of the molecular alterations in the aging brain [83], which may reduce global healthcare costs, benefit societies, and revolutionize healthcare practices universally.

## 3. Multivariate Influences and Variability in Age-Related Cognitive Decline

Heterogeneous changes across various functional domains characterize aging. While some individuals retain cognitive resilience and demonstrate the acquisition of complex decision-making and adaptive thinking, often referred to as ‘wisdom’, others experience significant declines, particularly in psychomotor processing speed and other cognitive functions [84,85,86,87]. One of the fundamental questions in the neuroscience of aging is explaining why some individuals decline more rapidly than others during healthy aging [88]. This variability leads many to desire to live only as long as they can preserve their functional autonomy [13,89]. In keeping with this phenomenon, there is increasing anxiety about the rising prevalence of dementia in the general population [90,91].

To uncover the underlying causes of age-related cognitive decline, previous studies have investigated a range of multivariate factors and have advocated for large-scale studies to assess the impact of cognitive aging on daily life activities [92,93]. Research exploring variability across different age groups suggests that both inter-individual variability (differences among individuals) and intra-individual variability (differences within an individual across various tasks) may predict age-related cognitive decline [94,95]. Compared to younger adults, older adults demonstrate greater inter-individual variability in neuropsychological test performance [96,97,98,99,100].

Normative cognitive data from the Cambridge Neuropsychological Test Automated Battery (CANTAB), encompassing tests on reaction time, spatial working memory, and paired-associate learning, reveal multiple trajectories of cognitive aging. This variability is particularly evident in episodic memory, as indicated by paired-associate learning results [101]. Older adults exhibit more significant inter-individual variability in neuropsychological test performance compared to younger individuals [97,98,99,100].

Numerous studies have identified various modifiable factors that influence cognitive outcomes in old age [102,103,104,105]. Enhanced cognitive performance in older adults is associated with higher levels of social engagement [106,107,108], greater educational attainment [109,110,111,112,113,114], and increased participation in enrichment activities, including physical exercise [25,106,113,115], reading, playing games, or engaging in hobbies [106,111,113].

Furthermore, memory decline associated with normal or pathological aging is exacerbated by institutionalization [116,117,118]. Institutional environments often provide limited sensory-motor and cognitive stimulation, social interaction, and physical activity, all contributing to a sedentary lifestyle [116,117]. Elderly individuals in long-term care facilities who received multisensory, motor, and cognitive stimulation twice a week for six months—including language and memory exercises, as well as visual, olfactory, auditory, and ludic stimulation such as music, singing, and dance—demonstrated improved cognitive performance. These findings highlight that long-term care facilities’ impoverished environments and sedentary lifestyles negatively affect cognitive performance, while cognitive stimulation leads to significant improvements [119]. However, four months after the program ended, a return to a sedentary lifestyle in long-term care facilities resulted in a substantial loss of the cognitive functions that had been regained [120]. In poorer countries, where significant inequalities negatively impact education, healthy elderly individuals with less education exhibited the poorest performance on CANTAB tests, particularly in sustained visual attention, reaction time, spatial working memory, and episodic memory [112].

Supporting the view is a significant volume of data from hippocampal studies demonstrating that aging affects the integrity of learning and memory consolidation, affective behaviors, and mood regulation. This is associated with increased oxidative stress and neuroinflammation, disturbances in intracellular signaling, alterations in gene expression, as well as shifts in synaptic plasticity and neurogenesis [121,122,123,124,125], all of which are exacerbated by a sedentary lifestyle [126]. Longitudinal research has demonstrated that physical activity supports the preservation of neuronal structural integrity and brain volume (the “Hardware”), while cognitive training for executive functions improves neural circuits, functions, and plasticity (the “Software”) [127].

The individual variability in age-related cognitive decline, both in extent and rate and its association with physical activity has been extensively explored [57,128]. Numerous studies provide compelling evidence that cognitive reserve—the brain’s resilience to neuropathological damage—can be enhanced through physical activity. Longitudinal studies [106], intervention studies [33,129] and meta-analyses [24,130] indicate that exercise is associated with greater cognitive reserve. Cognitive reserve is defined as the accumulation of neural resources that mitigates the effects of age-related cognitive decline [131,132,133,134,135,136], and it has been proposed as a critical factor for cognitive variability in later life [136].

Research shows that caloric restriction (CR) plays a significant role in cognitive health, reducing biomarkers of cellular senescence in humans [137,138]. The multicenter Comprehensive Assessment of Long-term Effects of Reducing Intake of Energy (CALERIE) study was the first randomized controlled trial on calorie restriction (CR), employing an innovative 25% caloric reduction design and methodology [139] on middle-aged, non-obese adults, providing essential insights into how caloric restriction can enhance cognitive function and overall health. The CALERIE^TM^ 2 trials analysis [140] showed long-term caloric restriction effects on human physiological, psychological, and behavioral outcomes [141] as well as on telomere length in healthy adults [142]. The CALERIE study also showed that calorie restriction affects the transcription of genes related to stress response and longevity in human muscle tissue [143]. Physical activity provides a robust physiological stimulus that induces molecular changes translated into multiple tissue crosstalk, improving homeostasis and decreasing the risk of premature mortality [144]. Reducing sedentary behavior and increasing physical activity have been shown to significantly improve metabolic and physical function, particularly in older adults engaged in exercise training programs [145]. Exercise interacts with dietary factors and has neurocognitive benefits on brain functioning [146,147,148]. Valuable insights were gained into how caloric restriction, combined with diet and physical activity, can improve cognitive function and overall health [149,150,151,152].

Additionally, studies from Wake Forest University [153] and the Exercise and Nutritional Interventions for Neurocognitive Health Enhancement (ENLIGHTEN Randomized Clinical Trial) [154] have further explored the synergistic effects of dietary interventions and physical activity, showing improvements in cognitive function across diverse populations. Insights from the “Blue Zones”—regions renowned for the exceptional longevity of their populations, including Okinawa in Japan, Sardinia in Italy, Loma Linda in California, United States, Nicoya in Costa Rica, and Icaria in Greece—highlight the influence of lifestyle factors on health and longevity [155,156]. Despite the geographical differences among Blue Zones—some being islands, others hilly or remote, and some surprisingly urban—these areas share key commonalities [157]. Figure 1 is a geographic map indicating the five Blue Zones. While the concept of Blue Zones has gained widespread attention for highlighting regions where people reportedly live longer, some experts question its validity [158].

Research indicates that the residents of these regions share common traits, such as daily physical activity, a predominantly plant-based diet, and strong social networks, all of which promote longevity and reduce the risk of chronic diseases [159,160]. Specifically, the Okinawans, often cited as a model for healthy aging, engage in lifelong physical activity through traditional practices like gardening, walking, and martial arts, which are closely linked to better cardiovascular health, enhanced cognitive function, and overall vitality in later life [161,162]. Research indicates that physically active lifestyles are linked to lower levels of inflammation, enhanced metabolic function, and increased resilience against age-related diseases, which contribute to a longer lifespan and increased numbers of years of ‘good health’ [163,164]. This evidence highlights the critical role of integrating regular physical activity and diet in aging populations to promote longevity and quality of life.

These findings, along with previous seminal studies, e.g., [32,33], underscore the critical importance of integrating dietary changes and physical activity as a combined strategy for enhancing cognitive outcomes. Additionally, the circulating microRNA profile of long-lived Okinawans revealed the upregulation of five specific microRNAs, which were exclusively present in both male and female nonagenarians with the longevity genotype [165].

In contrast, overweight and obesity are strongly linked to cognitive decline through several mechanisms, including chronic inflammation, insulin resistance, and alterations in brain structure, particularly in regions such as the hippocampus and prefrontal cortex [166,167]. Increased fat mass, particularly visceral fat, has been associated with higher levels of pro-inflammatory cytokines, contributing to neurodegenerative processes [168,169]. High-fat diets increase oxidative stress, cellular inflammatory response, and cognitive dysfunction [170]. In contrast, physical activity is crucial in mitigating these effects by improving cardiorespiratory fitness and favorably altering body composition through reductions in fat mass and increases in fat-free mass [171]. Cardiorespiratory fitness has been identified as a strong predictor of mortality risk, even from non-cardiovascular and non-cancer causes, underscoring the importance of promoting fitness as part of precision lifestyle recommendations [172]. Cardiorespiratory fitness has been linked to improved cognitive function, especially in areas such as executive function and memory, due to its role in increasing cerebral blood flow and supporting neuroplasticity [33,173].

Additionally, co-morbidities such as type 2 diabetes [174], cardiovascular disease [175], and stroke [176] further exacerbate cognitive decline in the elderly. Type 2 diabetes, for example, is associated with insulin resistance in the brain, which impairs glucose metabolism and accelerates amyloid plaque deposition—a hallmark of Alzheimer’s disease [177,178,179]. Cardiovascular disease and stroke contribute to vascular dementia through ischemic injury and hypoperfusion of brain tissue, leading to significant cognitive deficits [180]. Addressing these co-morbidities through lifestyle interventions that include weight management, physical activity, and control of metabolic risk factors is therefore critical for mitigating cognitive decline in aging populations [129,181,182,183].

Adequate sleep is crucial for cognitive function, particularly in processes such as memory consolidation, learning, and overall brain plasticity [184]. Sleep quality has been found to influence body composition, especially in older women, highlighting the importance of sleep interventions in maintaining metabolic health [185]. Chronic sleep deprivation or poor sleep quality has been linked to accelerated cognitive decline [186], an elevated risk of neurodegenerative diseases, including Alzheimer’s, along with structural changes in the brain, especially in the hippocampus and prefrontal cortex, which are crucial for memory and executive function [187,188]. Sleep disturbances can also exacerbate inflammation and oxidative stress, both of which are detrimental to brain health [188,189].

Moreover, sleep is intricately connected to other lifestyle factors, such as physical activity and sedentary behavior [190,191]. Regular physical activity has been shown to improve sleep quality by increasing the duration of slow-wave sleep, which is the most restorative sleep stage, and by reducing the time it takes to fall asleep [190,192]. In contrast, prolonged sedentary behavior, such as excessive screen time or sitting, can disrupt circadian rhythms, leading to poor sleep patterns and further impairing cognitive function [193,194,195]. Additionally, inadequate sleep is associated with metabolic dysregulation, including insulin resistance and increased appetite, which can contribute to obesity—a known risk factor for cognitive decline [196,197]. Given the strong bidirectional relationship between sleep, cognitive health, and other lifestyle factors, promoting good sleep hygiene alongside physical activity and minimizing sedentary time is essential for preserving cognitive function in aging populations [198,199]. In addition, the gut microbiome may affect sleep quality and health, eventually requiring support to provide appropriate dietary fiber, unsaturated fatty acids, and polyphenols along with time and spacing to guarantee microbiota’s capacity to produce essential metabolites for quality sleep and these include short-chain fatty acids, tryptophan, serotonin, melatonin, and gamma-aminobutyric acid [200].

## 4. Molecular Determinants of Cognitive Reserve, Resilience, and Age-Related Cognitive Decline

Understanding the molecular determinants of cognitive reserve and resilience is crucial for elucidating the variability in age-related cognitive decline [118]. The Bronx Aging Study of 1988 demonstrated that individuals with higher educational levels exhibited fewer clinical signs of dementia despite having the same degree of neuropathological changes as those with lower education [119]. In 1994, a landmark study by Stern supported this finding and introduced the concept of cognitive reserve [120].

In 2018, the concept of cognitive reserve was refined to encompass the adaptability of cognitive processes, linking the differential susceptibility of cognitive abilities and daily functioning to brain aging, pathology, or injury. Concurrently, resilience was defined as the capacity of the brain to maintain cognition and function despite aging and disease [118]. However, the cellular and molecular basis underlying the heterogeneous cognitive disabilities observed in the aged population remains poorly understood. Understanding these pathways is essential for providing insights into the biological basis of cognitive decline and identifying potential therapeutic targets [121].

Multiple factors influence the imbalance in homeostasis during senescence, and these include DNA instability, telomere attrition, epigenetic changes, loss of proteostasis, mitochondrial dysfunction, cellular senescence, nutrient dysregulation, stem cell exhaustion, and altered intercellular communication (Figure 2) [93,201,202,203]. These interconnected hallmarks of aging have significantly advanced our understanding of the biological processes underlying aging.

Over the past decade, significant advancements in our understanding of aging have identified additional hallmarks contributing to the mechanisms underlying age-related diseases. These newly recognized factors include impaired macroautophagy, which disrupts the cellular process of degrading and recycling damaged components; age-associated dysbiosis, characterized by an imbalance in the gut microbiota that affects immune function and metabolic health; and altered mechanical properties of tissues, such as decreased elasticity and increased stiffness, which compromise organ function. Furthermore, splicing dysregulation, involving errors in the processing of pre-mRNA, has been implicated in the loss of proteostasis and the accumulation of dysfunctional proteins, while chronic inflammation, often referred to as “inflammaging” [204], perpetuates tissue damage and accelerates aging across multiple systems [205]. These emerging hallmarks underscore the interdependence and complexity of aging processes, necessitating a paradigm shift in understanding aging—not merely as a collection of isolated events but as an intricate network of interconnected pathways. Recognizing this interconnectedness is crucial for developing more effective interventions to target the multifaceted nature of aging and age-related diseases.

Genomic instability results from the loss of efficiency of DNA repair mechanisms with age, accumulating genomic damage, and the ectopic presence of DNA, such as cytosolic [206]. As we age, DNA damage increases, and its degradation is incomplete, inducing an immune reaction exacerbating chronic low-grade inflammation in aging. Therefore, strategies that either reduce the induction of cytoplasmic DNA or enhance its clearance are becoming attractive therapeutic targets [207]. Telomeres, i.e., the protective ends of linear chromosomes, shorten throughout an individual’s lifespan, and critically short telomeres induce genomic instability, apoptosis, or cell senescence, leading to aging and age-associated diseases [208]. In most mammalian somatic cells, telomerase is not expressed, preventing replicative DNA polymerases from fully replicating the telomeric regions of eukaryotic DNA. This incomplete replication results in DNA damage at the chromosome ends, contributing to aging and age-related diseases [208,209]. Epigenetic dysregulation is another critical hallmark of aging [210]. Fundamental epigenetic changes contributing to the aging process include alterations in DNA methylation patterns [211], abnormal post-translational modification of histones [212,213], aberrant chromatin remodeling [214,215], and deregulated function of non-coding RNAs [216,217,218]. These non-coding RNAs—including microRNAs, long non-coding RNAs, and circular RNAs—play crucial roles in learning, memory, and adaptive immunity [219].

A complex proteostasis network in healthy cells—comprising molecular chaperones, proteolytic machinery, and their regulators—maintains proteins’ levels, structure, and function, ensuring overall protein homeostasis (proteostasis). However, as cells age, inevitable endogenous and external stresses make it increasingly difficult to sustain this balance [220]. Disruption of proteostasis results in the buildup of misfolded, oxidized, and/or ubiquitin-tagged proteins, which can form intracellular inclusions or extracellular amyloid plaques [221,222]. This decline in the proteostasis network compromises the integrity of the proteome, contributing to cellular dysfunction and age-related diseases (Figure 2).

## 5. Neuroinflammation, Exercise, and Cognitive Variability in Old Age

Recent studies have extensively explored the relationship between neuroinflammation and aging, significantly advancing our understanding of age-related cognitive decline mechanisms [223,224]. Aging seems to be associated with the upregulation of the inflammatory processes in the brain [225], including the activation of immune-related cells, such as microglia and astrocytes within the CNS, which has been identified as a critical factor in the progression of aging and cognitive impairments [226,227]. Chronic inflammation promotes cell senescence and immunosenescence that leads to impaired clearance mechanisms [228]. At the same time, the accumulation of damaged DNA and senescent cells and the factors they release promote neuroinflammation in the aging brain [217]. Senescent cells were shown to become more abundant in mice with aging, and their depletion mitigated neuroinflammation and delayed cognitive impairment [229]. Neuroinflammation was recently demonstrated to have a causative role in structural and functional connectivity impairment and to favor Alzheimer’s disease progression [230]. Heterogeneity in aging trajectories is associated with different lifestyle behaviors, where reduced physical and cognitive activity is linked to faster rates of decline [231,232,233].

Translational research, bridging animal and in vitro studies with human postmortem data, has demonstrated a causal link between physical activity and the maintenance of microglial homeostasis [234]. When physical activity was objectively monitored using accelerometer-based actigraphy, studies revealed its protective effects on age-related brain structure and function [235]. These studies showed that physical activity negatively correlates with Lewy body disease and is positively associated with Alzheimer’s disease burdens. Moreover, in older adults, particularly nonagenarians, the proportion of activated microglia in sampled brain areas was lower in those physically active, associated with greater cognitive resilience [234]. In addition, late-life physical activity was linked with markers of synaptic integrity in brain tissue, further emphasizing its role in preserving cognitive function in the aging population [234].

Building on the relationship between physical activity and cognitive resilience, a postmortem study conducted as part of the Memory and Aging Project at the Rush Alzheimer’s Disease Center investigated the connection between neuropathological changes and various lifestyle factors in older adults. A recent study by Paolillo et al. [158] revealed that certain lifestyle behaviors can help maintain cognitive stability, even in significant neuropathological changes. The findings suggest that engaging in various lifestyle activities, including environmental enrichment and social interactions, may slow cognitive decline and delay the progression of clinical symptoms in older adults. The results emphasize the potential of lifestyle modifications to preserve mental function, particularly in the oldest age groups, despite underlying neuropathological conditions [236]. Interestingly, the effect of a healthy lifestyle behavior may have more benefits in women than in men [237,238,239].

Studies in mouse models of Alzheimer’s disease have also demonstrated that physical activity attenuates the expression of pro-inflammatory markers, cognitive deficits, astrocytosis, brain amyloid-beta deposition, and disease progression [240,241]. Previous studies have demonstrated that aged mice and rats housed in standard laboratory cages exhibited poorer performance in learning and memory tasks than those in enriched environments [242,243,244,245,246,247,248,249,250].

In mice, cognitive training attenuated age-related decline in visual discrimination and behavioral flexibility [251]. Moreover, enriched-housed mice showed enhanced neural plasticity, improved memory formation in learning tasks, and molecular and neural structural changes in the prefrontal cortex [252]. Pioneer publications demonstrated that environmental enrichment and voluntary exercise were beneficial to neuronal and neuroimmune functions in both young and aged individuals [253,254,255,256,257,258,259,260,261,262]. Figure 3 synthesizes the current view of the beneficial effects of an active lifestyle on spatial learning and memory, glial cell morphology, and neuroinflammation.

It has been suggested that glial cells are the first populations to exhibit age-related changes within the brain [263]. Interestingly, single-cell transcriptomics of the aging glia identified a shared age-induced molecular signature across all major glial cell types related to mitochondrial dysfunction, loss of proteostasis, and cellular senescence [264]. Glial cell senescence contributes to synaptic dysfunction, neuroinflammation, and impaired neurogenesis [229]. Astrocytes, microglia, and neuron-glia antigen 2-expressing glial cells (NG2 glia, a specific glia cell type) participate in the regulation of synaptic functions and plasticity [185]; their dysregulation by aging-induced cell senescence may contribute to cognitive impairment and synaptic dysfunction and, eventually, to the progression of Alzheimer’s disease pathology [265,266]. Physical exercise has been demonstrated to increase the astrocyte coverage of cerebral blood vessels and to counteract cognitive decline in a rat model of chronic cerebral hypoperfusion [267], as well as to maintain the homeostasis of cortical microcircuits by reshaping microglial cells in the mutant human TDP-43 proteinopathy mouse model [268].

Recent research showed that astrocyte dystrophy in physiological aging favors glutamate spillover and parallels impaired synaptic plasticity in the C57BL/6 male mice [269]. Accumulating evidence indicates that exercise enhances synaptic and cerebrovascular plasticity and the density of dendritic spines, improving neuroplasticity [270]. Recently, voluntary running exercise for four months was revealed to ameliorate spatial learning and memory abilities, as well as to increase the total number of dendritic spines and synaptic dynamics in the hippocampus of the APP/PS1 mice, an Alzheimer’s disease model [271]. Surprisingly, running exercise reduced the expression of advanced glycation end products (AGEs), the receptors for advanced glycation end products (RAGE), complement component 1q (C1q), and complement component 3 (C3) in the hippocampus, and the number of Iba1+ microglia.

Physical exercise also enhanced the expression of brain-derived neurotrophic factor (BDNF) in astrocytes, further improving hippocampal neuroplasticity, a critical factor in cognitive function and memory formation [272]. By upregulating lactate levels, exercise further contributed to neuronal energy supply by promoting the astrocyte–neuron lactate shuttle (ANLS) [273]. Importantly, lactate produced by astrocytes has been found to play a crucial role in long-term potentiation (LTP) at neural synapses, which is fundamental for synaptic strength, memory formation, and neurorehabilitation [274,275].

Changes in astrocyte morphology by exercise were found to be region-specific and may be supportive of synaptic integrity and synaptogenesis [275]. Moreover, voluntary wheel running in the aged mice triggered the upregulation of aquaporin 4 (AQP4) in both the cortex and the hippocampus and end-feet polarization. AQP4 is a critical component of the lymphatic–glymphatic system, which facilitates the exchange of cerebrospinal fluid and interstitial fluid, highlighting the role of exercise in promoting brain health through astrocyte function [276]. Upregulation of astrocyte gene expression in glial fibrillary acidic protein (GFAP), thrombospondin 2 (Thbs2), leukemia inhibitory factor (Lif), and interleukin 6 (IL-6) by physical activity [275] deserve further studies to explore whether it helps in immune responses, such as those against COVID-19 [277].

By releasing diverse signaling molecules, microglia, and astrocytes establish autocrine feedback and crosstalk that regulate cell phenotypes and response to challenges, which are fundamental to neural function and dysfunction [278,279]. While astrocyte modifications by aging predominantly relate to genes associated with synaptic transmission, the genes upregulated in microglia are associated with the inflammatory response [280]. Age-related release of neuroinflammatory cytokines from the activated microglia and astrocytes disrupts synaptic plasticity, exacerbates neuronal damage, reduces neurogenesis, and ultimately leads to cognitive deficits [201,226,281]. Conversely, human studies have shown that exercise reduces neuroinflammation and the risk of developing dementia [56,232].

Microglia in the aging context show a reduced ability to damage and pathological cues [282] and exhibit the downregulation of genes involved in cell adhesion, motility, and phagocytosis [283,284,285]. Moreover, age-related synapse loss and cognitive decline may be linked to increased pruning of brain cell connections by microglia by mechanisms involving trogocytosis instead of phagocytosis [286]. The production of leukotrienes (small lipid mediators of inflammation) by microglia during aging and neurodegeneration is argued to be pivotal in age-related cognitive decline and chronic neurodegenerative diseases [287,288,289,290]. In an exploratory study, leukotriene signaling was described as a molecular correlate for cognitive heterogeneity in aging [291]. As microglia are the primary producers of leukotrienes in the brain, the use of leukotriene receptor antagonists has been demonstrated to reduce neuroinflammation, as well as to promote neurogenesis, and restore cognition in aged rats [292].

Physical exercise has been demonstrated to counterbalance the impact of aging on astrocyte–microglia communication, influencing both neuroinflammation and neural plasticity [290] and ensuring an optimal environment for brain plasticity and cognitive function [273,293]. Consistent physical activity has been shown to lower the expression of neuroinflammatory markers, boost hippocampal neurogenesis and synaptic plasticity, and stimulate the production of anti-inflammatory cytokines [294,295,296]. The anti-inflammatory effects of exercise are partially mediated by the increased expression of neurotrophic factors, such as brain-derived neurotrophic factor (BDNF), which promotes neuronal survival and enhances synaptic plasticity [297].

miRNAs are essential modulators of microglial phenotype with age-dependent specific subsets and refinement by epigenetic mechanisms that influence microglial functions and subtypes [298,299]. miR-29a-3p and miR-132-3p were recently identified as the most significant miRNAs associated with cognitive trajectory [300]. Interestingly, exercise training was shown to cause long-lasting changes in the expression of miRNAs [301] and cognitive improvement [302]. miR-129-5p, miR-192-5p, miR-15b-5p, miR-148b-3p, miR-130a-3p, and miR-132 sorted out as the most related to aerobic exercise training [303], while miR-409 and miR-501 were identified as the most increased in the hippocampus upon exercise and as being correlated with cognition [304].

Small extracellular vesicles are argued to be the principal carriers of miRNAs, which can be transmitted to target cells by fusion, receptor–ligand interaction, endocytosis, and phagocytosis [305]. Extracellular vesicle-derived miRNAs are key in cell-to-cell communication [306] and the gut–brain–microbiota axis [307]. Gut microbiota communicates with microglia through secreted metabolites, determining morphological and functional changes in microglia that are associated with neuroinflammation and age-related cognitive decline [308]. Indeed, new insights indicate that changes in gut microbiota composition, microbial metabolites, neurotransmitters, astrocyte reactivity, and microglial activation subtypes in cognitive decline and neurodegenerative diseases are interrelated during aging [309,310]. Several studies support the view that exercise modifies the gut microbiota with modulation of human moods and behaviors, triggering positive health effects [311,312]. Intriguingly, it also stimulates motivation for exercise [313,314]. In summary, manipulating the microbiota–gut–brain axis holds promise as an accessible route to modulate glial functions indirectly [315].

## 6. Clinical Significance and Lifestyle Prescription

Valuable insights for clinicians and practitioners aiming to incorporate precision medicine, combining individual gene variation with the environment into individual lifestyle prescriptions, have recently emerged [316,317,318]. For example, not all exercise regimens are universally effective in all individuals and should not be treated as a universal prescription [316]. The emerging field of precision medicine emphasizes tailoring interventions based on individual genetic, environmental, and lifestyle factors. Data suggest clinicians should consider personalized lifestyle recommendations based on the patient’s metabolic profile, physical activity levels, and genetic predisposition in this context. Evidence suggests that racial and ethnic differences influence how individuals respond to physical activity and cardiorespiratory fitness interventions, necessitating a more tailored approach in clinical practice [172]. For example, patients exhibiting metabolic dysregulation or prediabetic tendencies may benefit from personalized dietary recommendations, such as following a Mediterranean diet, linked to better glycemic control and improved cardiovascular health in high-risk populations [319]. Caloric restriction, beyond weight loss, has also been linked to improvements in health span and lifespan, suggesting that diet-based interventions can profoundly impact overall health [320]. Additionally, encouraging physical activity—such as 150 min of moderate aerobic exercise per week—improves insulin sensitivity and overall metabolic health [321,322,323,324]. Physical activity is essential in reducing glycemic variability, a critical factor in managing metabolic conditions such as diabetes [325]. Even non-exercise-based estimates of cardiorespiratory fitness predict type 2 diabetes onset, reinforcing the importance of incorporating fitness assessments into routine care [326]. Furthermore, incorporating behavioral interventions based on individual genetic and environmental backgrounds could significantly enhance the long-term success of lifestyle modifications. Evidence suggests that sleep patterns [327,328], stress management [329,330,331], and even gut microbiota composition play pivotal roles in metabolic health [332,333,334,335]. Clinicians should consider incorporating strategies that promote sleep hygiene and stress reduction, such as mindfulness-based stress reduction (MBSR) or cognitive-behavioral therapy (CBT), which have been shown to mitigate stress-related metabolic changes [336,337]. In summary, by integrating personalized lifestyle interventions rooted in precision medicine principles, healthcare providers can optimize the management of metabolic disorders and reduce the risk of associated complications.

## 7. Concluding Remarks

In modern societies, sedentary behavior has significantly contributed to cognitive impairments among aging populations. This review underscores the necessity of adopting a multifaceted approach to mitigate these effects, emphasizing the importance of integrating physical exercise with cognitive, multisensory, and motor stimulation. Such interventions have consistently demonstrated their ability to enhance cognitive performance and delay the progression of severe neurodegenerative conditions, including Alzheimer’s disease.

Understanding the molecular, cellular, and systemic mechanisms underlying cognitive decline is crucial for developing effective therapeutic strategies. This integrative review has dissected the complex interplay of these mechanisms, highlighting the critical role of an active lifestyle in counteracting cognitive deterioration. By identifying key molecular signatures, this review also elucidated the therapeutic benefits of non-pharmacological interventions, drawing on a synthesis of empirical evidence from both experimental models and human studies. We have particularly sought to highlight key aging hallmarks—such as genomic instability, telomere attrition, and neuroinflammation—to emphasize the need for comprehensive and personalized approaches to promote cognitive resilience and healthy aging. Tackling these interconnected factors with multitargeted therapies and non-pharmacological interventions can significantly reduce cognitive decline, improving the quality of life for aging populations.

Future research should further explore these complex interactions and develop targeted and personalized therapies that address the multifactorial nature of cognitive decline. By fostering environments that encourage physical, cognitive, and social engagement, we can better support cognitive health and significantly improve the well-being of older adults at a relatively low cost. This integrative approach may delay cognitive decline and transform the aging experience, enabling individuals to maintain a higher quality of life into old age while reducing the societal burden.

## Figures and Tables

**Figure 1 ijms-25-10757-f001:**
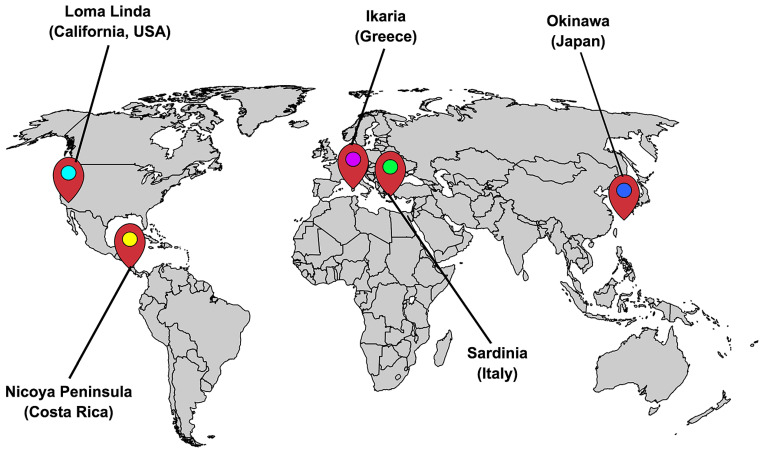
Blue Zones: Okinawa (Japan), Sardinia (Italy), Ikaria (Greece), Nicoya Peninsula (Costa Rica), and Loma Linda (California, United States). Each location is marked in red and labeled for clarity.

**Figure 2 ijms-25-10757-f002:**
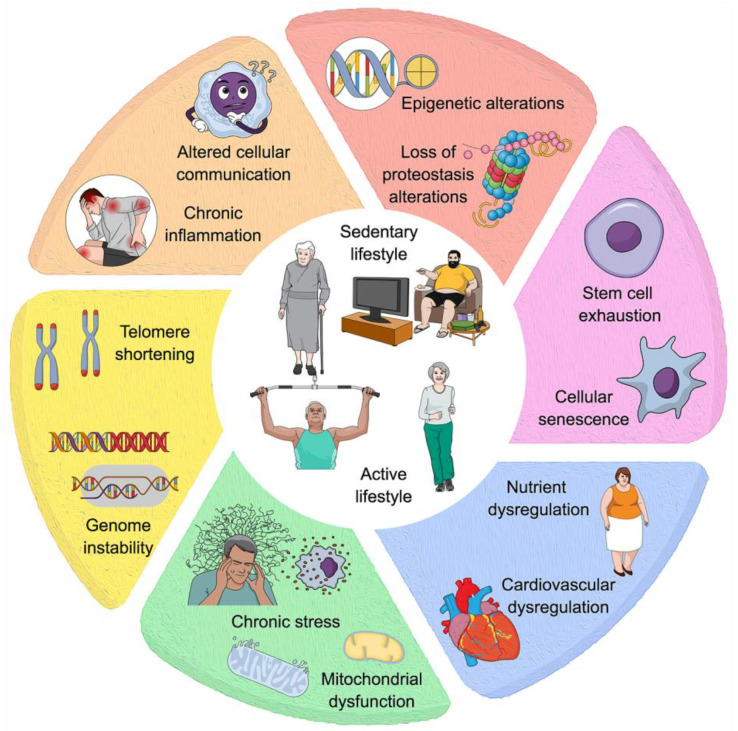
Sedentary versus active lifestyles impact choices on several hallmarks of aging and cellular health. The outer sections show different biological processes and cellular alterations influenced by these lifestyle choices. Telomere shortening and genome instability (yellow section); chronic stress and mitochondrial dysfunction (green section); nutrient dysregulation and cardiovascular dysfunction (blue section); stem cell exhaustion and cellular senescence (pink section); epigenetic alterations and proteostasis (orange section); chronic inflammation and altered cellular communication (vanilla color section). Inactivity induces chronic inflammation and impairs cellular communication, exacerbating tissue damage and contributing to the onset of age-related chronic diseases. In contrast, an active lifestyle mitigates these adverse effects by maintaining cellular function, reducing stress, improving proteostasis, and promoting cardiovascular health, which can collectively slow down the aging process.

**Figure 3 ijms-25-10757-f003:**
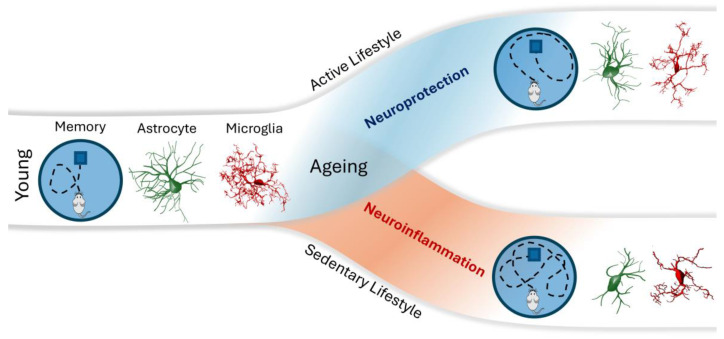
Impact of aging on memory, astrocytes, and microglia, highlighting the contrasting effects of an active versus sedentary lifestyle. On the left, the “Young” state is depicted, where memory function, astrocytes (green), and microglia (red) are in a healthy, balanced state. As aging progresses, the path diverges based on lifestyle: Active Lifestyle (upper blue path): An active lifestyle promotes neuroprotection, helping preserve memory function, maintain healthy astrocyte and microglial function, and mitigate the detrimental effects of aging. Sedentary Lifestyle (lower red path): A sedentary lifestyle leads to neuroinflammation, impairing memory, dysregulating astrocyte function, and activating microglia in ways that exacerbate neurodegenerative processes.

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
