# Peer review of "The Hidden Dangers of Sedentary Living: Insights into Molecular, Cellular, and Systemic Mechanisms"

_ijms, 2024, doi:10.3390/ijms251910757_

Round 1
Reviewer 1 Report
Comments and Suggestions for Authors
This article is a review of the impact of a sedentary lifestyle on cognitive decline in an aging society. However, it falls short in several areas.
1. In Introduction, the authors need to provide more comprehensive literature reviews, particularly focusing on studies concerning sedentary behavior and cognitive decline, as well as limitations of current researches.
2. The formatting of this article significantly diverts from the standard MDPI guidelines.
3.Section 2 provides an inadequate explanation of the relationship between cognitive decline and risk factors in medical imaging.
Ref:
Montagnese M, Rittman T. Bridging modifiable risk factors and cognitive decline: the mediating role of brain age. Lancet Healthy Longev. 2024 Apr;5(4):e243-e244.
Lin, L.; Xiong, M.; Jin, Y.; Kang, W.; Wu, S.; Sun, S.; Fu, Z. Quantifying Brain and Cognitive Maintenance as Key Indicators for Sustainable Cognitive Aging: Insights from the UK Biobank. Sustainability 2023, 15, 9620.
4.Section 2 is somewhat diffused and lacks enough focus on physical activity’s relationship to cognition.
5. The title does not encompass cognitive decline. This should be revised accordingly.
6.The conclusion section could be more detailed in discussing how the findings of this study could be translated into practical interventions or policy recommendations.
7.The iThenticate report indicates that the paper exhibits substantial textual overlap with previously published work, it is essential to thoroughly revise the content.
Author Response
Author's Reply to the Review Report
Reviewer 1 Round 1
This article is a review of the impact of a sedentary lifestyle on cognitive decline in an aging society. However, it falls short in several areas.
- In Introduction, the authors need to provide more comprehensive literature reviews, particularly focusing on studies concerning sedentary behavior and cognitive decline, as well as limitations of current researches.
Reply: Thank you for your insightful feedback. We have rewritten the Introduction to provide a more comprehensive literature review, focusing on studies addressing the link between sedentary behavior and cognitive decline. We have also discussed the limitations of current research in these areas, highlighting gaps in the literature and the need for further studies to explore the long-term effects of sedentary behavior on cognitive health. We believe these revisions provide a stronger foundation for the study and align with the Reviewer’s recommendations.
- The formatting of this article significantly diverts from the standard MDPI guidelines.
Reply: Thank you for pointing this out. We have thoroughly reviewed and revised the manuscript to ensure it adheres to the MDPI formatting guidelines. All sections, including headings, citations, references, and overall layout, have been adjusted to comply with the required standards. We appreciate your attention to detail and believe the manuscript fully aligns with MDPI’s formatting requirements.
3.Section 2 provides an inadequate explanation of the relationship between cognitive decline and risk factors in medical imaging.
Ref: Montagnese M, Rittman T. Bridging modifiable risk factors and cognitive decline: the mediating role of brain age. Lancet Healthy Longev. 2024 Apr;5(4):e243-e244.
Lin, L.; Xiong, M.; Jin, Y.; Kang, W.; Wu, S.; Sun, S.; Fu, Z. Quantifying Brain and Cognitive Maintenance as Key Indicators for Sustainable Cognitive Aging: Insights from the UK Biobank. Sustainability 2023, 15, 9620.
Reply: Thank you for your valuable feedback. In response to your comment regarding the inadequate explanation of the relationship between cognitive decline and risk factors in medical imaging in Section 2, we have expanded this section to provide a more detailed discussion. Specifically, we have added two paragraphs highlighting the role of imaging biomarkers in revealing the cumulative effects of modifiable risk factors such as hypertension, diabetes, and smoking on brain aging. We integrated findings from Montagnese and Rittman (2024) and Lin et al. (2023), emphasizing how brain age serves as a mediator between these risk factors and cognitive decline. Additionally, we have discussed how imaging markers, including cortical thickness and white matter integrity, are influenced by these factors and contribute to understanding and tracking cognitive aging.
These additions aim to address the gap in the original manuscript by providing a more comprehensive explanation of how medical imaging techniques correlate with both risk factors and cognitive performance. Thank you once again for your helpful suggestions, and we hope these revisions meet your expectations.
- Section 2 is somewhat diffused and lacks enough focus on physical activity’s relationship to cognition.
Reply: Thank you for your valuable feedback. In response to your comment regarding the lack of focus on physical activity’s relationship to cognition in Section 2, we revised this section to provide a more focused discussion. We emphasized the substantial evidence linking regular physical activity to enhanced cognitive function and a reduced risk of cognitive decline. Specifically, we highlighted how physical activity contributes to improved cardiovascular health, neuroplasticity, and the reduction of neuroinflammation—factors that are all closely associated with cognitive health as follows:
- “…Cognitive decline risk factors and aging models
Research over the past decade has advanced our understanding of the risk factors associated with age-related cognitive decline. Epidemiological studies have identified several modifiable and non-modifiable risk factors. Non-modifiable risk factors include age, genetics, and family history of dementia. Modifiable risk factors encompass lifestyle and health-related factors such as physical activity, diet, cardiovascular health, education, and social engagement [34, 36, 39]. Among these, physical activity is one of the most significant contributors to cognitive health. Regular physical exercise has been associated with a reduced risk of cognitive decline, primarily by improving cardiovascular health and promoting neurogenesis and synaptic plasticity [40-46]. Physical activity is also thought to mitigate neuroinflammation, which is linked to cognitive impairment in aging [47-50]
Studies have also highlighted the protective role of physical activity through its enhancement of cognitive reserve. More physically active individuals demonstrate greater resistance to cognitive decline, likely due to improved neurovascular health and increased brain plasticity [44]. This reserve allows individuals to maintain cognitive function despite age-related changes in the brain. The beneficial effects of physical activity extend beyond cardiovascular health; they directly impact the structural integrity of brain regions involved in memory and executive function, such as the hippocampus and prefrontal cortex [38]
This evidence underscores the value of incorporating regular physical activity into preventive strategies for dementia.
Recent advancements in understanding the biological processes contributing to cognitive decline have provided invaluable insights into the cellular and molecular alterations of aging [51]. Experimental models, particularly rodents, have facilitated controlled manipulation of genetic, environmental, and pharmacological factors, identifying critical molecular pathways involved in cognitive decline [52, 53]. Rodent studies have elucidated mechanisms such as synaptic plasticity, neurogenesis, and the impact of stress and diet on cognitive function [54]. Interestingly, it has been demonstrated that whole-body clearance of senescent cells can alleviate age-related increases in basal ‘brain inflammation’ and cognitive impairment in murine models [55], which highlights the role of the host response to the degenerative process as a contributor to the loss of cognitive function with age.
Human induced pluripotent stem cells and transdifferentiated cells from aged donors and patients as aging models in vitro may also help to unravel the cross-talk between aged and proliferative cells to understand how aging and disease develop [56] and how the responses can be reprogramed in a dish [57]. Additional human studies are necessary to provide the necessary validation and context to the experimental studies. Neuroimaging, genomics, and post-mortem brain analyses have revealed patterns of brain aging and cognitive decline, enabling the translation of findings from animal models to human conditions [58, 59]. Techniques such as magnetic resonance imaging (MRI) and positron emission tomography (PET) have been pivotal in visualizing structural and functional changes in the aging brain [30, 60, 61]. Medical imaging, particularly modalities like MRI and PET scans, has become increasingly valuable in identifying structural and functional changes in the brain associated with cognitive decline. Evidence suggests that imaging can reveal the cumulative effects of modifiable risk factors such as hypertension, diabetes, and smoking on brain aging. For example, Montagnese and Rittman (2024) [62] highlight that brain age, as measured by MRI, mediates these risk factors and cognitive decline, reinforcing the importance of early intervention strategies.
Additionally, as Lin et al. (2023) demonstrate [63], integrating imaging biomarkers with cognitive performance data—such as from the UK Biobank—provides a comprehensive framework for understanding sustainable cognitive aging. Imaging markers like cortical thickness and white matter integrity are directly affected by modifiable risk factors, which can accelerate brain aging and heighten the risk of cognitive decline. By quantifying these imaging markers, researchers can monitor brain health over time, making it a valuable tool for tracking brain maintenance and potentially mitigating the impact of these risk factors on cognition. These studies have identified brain regions particularly vulnerable to aging and have correlated these changes with cognitive performance. Advances in PET imaging have allowed the in vivo visualization of amyloid and tau pathology, directly linking molecular changes to cognitive decline [59, 64]. 18F fluoro-deoxy-glucose (FDG)-PET imaging has also been used to evaluate the alterations in cerebral glucose metabolism in MCI and was found to differentiate progressive MCI from stable MCI [65].
Genomic studies have identified genetic variants associated with an increased risk of cognitive decline and Alzheimer’s disease. Genome-wide association studies (GWAS) have uncovered numerous risk loci, including the well-known APOE ε4 allele, which is known to elevate the risk of Alzheimer’s disease significantly [66, 67]. These genetic findings have provided some insights into the molecular pathways involved in cognitive decline, highlighting the critical roles of cholesterol metabolism, immune response, and synaptic function [34] though the precise role of the ε4 allele has remained an enigma that demands to be solved. Other research that has examined telomere length has underscored its potential role in cognitive health, suggesting that the preservation of telomere length has cognitive benefits among aged individuals at risk of dementia [68].
Post-mortem brain analyses have confirmed and extended findings from neuroimaging and genomics studies, revealing the accumulation of amyloid plaques, neurofibrillary tangles, and other pathological features associated with Alzheimer’s disease [58, 69, 70]. These studies have also identified changes in neurotransmitter systems, synaptic density, and neuroinflammatory markers, providing a comprehensive picture of the molecular alterations in the aging brain [71]. Recently, artificial intelligence tools were proposed for early detection of neurocognitive impairments [72]…”
- The title does not encompass cognitive decline. This should be revised accordingly.
Reply: Thank you for your insightful feedback regarding the title. After considering your suggestion, we understand the need to ensure that the title accurately reflects the scope and impact of the study while remaining engaging and clear to the reader. The title "Sedentary life is an invisible trap: molecular, cellular, and systemic reasons to leave it behind" was chosen to emphasize the pervasive yet often overlooked consequences of a sedentary lifestyle, as well as to highlight the comprehensive molecular, cellular, and systemic mechanisms we explore. However, we are open to refining the title to better align with your expectations and suggestions. We propose the revised version: "The Hidden Dangers of Sedentary Living: Insights into Molecular, Cellular, and Systemic Mechanisms”.
- The conclusion section could be more detailed in discussing how the findings of this study could be translated into practical interventions or policy recommendations.
Reply: To address this issue, we have integrated a clinical significance and lifestyle prescription section into the manuscript as follows: “…
- Clinical Significance and Lifestyle Prescription
Valuable insights for clinicians and practitioners aiming to incorporate precision medicine, combining individual gene variation with the environment into individual lifestyle prescriptions, have recently emerged [319-321]. For example, not all exercise regimens are universally effective in all individuals and should not be treated as a universal prescription [319]. The emerging field of precision medicine emphasizes tailoring interventions based on individual genetic, environmental, and lifestyle factors. Data suggests clinicians should consider personalized lifestyle recommendations based on the patient’s metabolic profile, physical activity levels, and genetic predisposition in this context. Evidence suggests that racial and ethnic differences influence how individuals respond to physical activity and cardiorespiratory fitness interventions, necessitating a more tailored approach in clinical practice [174]. For example, patients exhibiting metabolic dysregulation or prediabetic tendencies may benefit from personalized dietary recommendations, such as following a Mediterranean diet, linked to better glycemic control and improved cardiovascular health in high-risk populations [322-324]. Caloric restriction, beyond weight loss, has also been linked to improvements in health span and lifespan, suggesting that diet-based interventions can profoundly impact overall health [325]. Additionally, encouraging physical activity — such as 150 minutes of moderate aerobic exercise per week — improves insulin sensitivity and overall metabolic health [326-329]. Physical activity is essential in reducing glycemic variability, a critical factor in managing metabolic conditions such as diabetes [330]. Even non-exercise-based estimates of cardiorespiratory fitness predict type 2 diabetes onset, reinforcing the importance of incorporating fitness assessments into routine care [331]. Furthermore, incorporating behavioral interventions based on individual genetic and environmental backgrounds could significantly enhance the long-term success of lifestyle modifications. Evidence suggests that sleep patterns [332, 333], stress management [334-336], and even gut microbiota composition play pivotal roles in metabolic health [337-340]. Clinicians should consider incorporating strategies that promote sleep hygiene and stress reduction, such as mindfulness-based stress reduction (MBSR) or cognitive-behavioral therapy (CBT), which have been shown to mitigate stress-related metabolic changes [341-343]. In summary, by integrating personalized lifestyle interventions rooted in precision medicine principles, healthcare providers can optimize the management of metabolic disorders and reduce the risk of associated complications…”
7.The iThenticate report indicates that the paper exhibits substantial textual overlap with previously published work, it is essential to thoroughly revise the content.
Reply: Thank you for your feedback and for bringing this to our attention. We take concerns about plagiarism seriously and have thoroughly revised the manuscript to ensure originality and avoid substantial textual overlap with previously published work. Where applicable, we have restructured sentences, employed passive voice, and paraphrased key concepts to reflect this study's unique contributions better while maintaining clarity and scientific rigor. Citations have been carefully reviewed to ensure proper attribution of ideas.
We hope that the revised manuscript now addresses your concerns appropriately.

Reviewer 2 Report
Comments and Suggestions for Authors
The correlation between physical activity and age-related cognitive decline has been the subject of extensive research, recent findings suggesting that physical activity may have a protective effect on cognitive function as individuals age. This manuscript supplied an overview of the relationship and key points to the mechanisms of correlation between these topics. There are several major and minor weaknesses in the rationale and research methods of this work. Below please find the review comments.
(1) Major comments
1. Rewrite the title. As a review manuscript discussing the correlation of age-related cognitive decline and lifestyle choices (sedentary?), it is crucial for the authors to revise the title to match the key words and the topics within this manuscript.
2. The authors should clarify these conceptions in the introduction section, including sedentary behavior, active lifestyle, physical exercise, healthy lifestyle behavior, and exercise, as they are not used uniquely in the following statements. And these vague concepts mean similar but with distinct definition, it is essential to clarify the types, strength, frequency, how to determine the normal (regular), sedentary, active lifestyle and over-active lifestyle, as well as the relations to age and age-related cognitive decline.
3. Meanwhile, the key words section also needs revising to make it match the topic and content, as there is little information about epigenetics, neuron-glia crosstalk, oxidative stress, protein aggregation, non-pharmacological and therapeutic interventions in the following statements.
4. It appears that line 495-618 is a complete repeat of line 372-494, no idea how it happened, please double check.
5. The figures should support the conclusions in a scientific manner, it requires clarity, relevance, and alignment with the data/statement presented. It is difficult to get the content of figure 1 and 2, please go through all the figures by filling in more details to make it alignment with the main text.
(2) Minor comments
1. Line 61-62, it is essential to add relevant references to supply sufficient evidence to support the statement here. Same issue applied to line 93. And please ensure that all references are carefully reviewed to verify that each citation is relevant to the corresponding statement and contributes to the overall conclusion.
2. Line 139 - 140, an uncompleted citation format appeared as this reference is not properly inserted. It is recommended to go through all the citation formats before submitting the manuscript to the system.
3. Line 355 - 356, this sentence needs revising to make it clear.
4. Please add more references to support the conclusion in line 362-363.
5. Line 337-338, “The heterogeneity in aging trajectories was demonstrated to be linked to different lifestyle behaviors and reduced physical and cognitive inactivity, to faster rates of decline in aging paths.”, I feel difficult to understand this sentence, please revise it and interpret the details.
6. There are issues with the English (syntax, grammar, etc.) throughout the manuscript. The authors should conduct a meticulous review of the whole manuscript to identify and correct any grammar and syntax issues. It includes missing/redundant space, italic issue, and grammar mistakes.
Comments on the Quality of English Language
Moderate editing of English language required.
Author Response
Reviewer 2 Round 1
Comments and Suggestions for Authors
The correlation between physical activity and age-related cognitive decline has been the subject of extensive research, recent findings suggesting that physical activity may have a protective effect on cognitive function as individuals age. This manuscript supplied an overview of the relationship and key points to the mechanisms of correlation between these topics. There are several major and minor weaknesses in the rationale and research methods of this work. Below please find the review comments.
(1) Major comments
- Rewrite the title. As a review manuscript discussing the correlation of age-related cognitive decline and lifestyle choices (sedentary?), it is crucial for the authors to revise the title to match the key words and the topics within this manuscript.
Reply: Thank you for your insightful feedback regarding the title. After considering your suggestion, we understand the need to ensure that the title accurately reflects the scope and impact of the study while remaining engaging and clear to the reader. The title "Sedentary life is an invisible trap: molecular, cellular, and systemic reasons to leave it behind" was chosen to emphasize the pervasive yet often overlooked consequences of a sedentary lifestyle, as well as to highlight the comprehensive molecular, cellular, and systemic mechanisms we explore. However, we are open to refining the title to better align with your expectations and suggestions. We propose the revised version: "The Hidden Dangers of Sedentary Living: Insights into Molecular, Cellular, and Systemic Mechanisms”.
- The authors should clarify these conceptions in the introduction section, including sedentary behavior, active lifestyle, physical exercise, healthy lifestyle behavior, and exercise, as they are not used uniquely in the following statements. And these vague concepts mean similar but with distinct definition, it is essential to clarify the types, strength, frequency, how to determine the normal (regular), sedentary, active lifestyle and over-active lifestyle, as well as the relations to age and age-related cognitive decline.
Reply: In response to the reviewer's comment, we have carefully revised the Introduction to clarify the concepts of sedentary behavior, active lifestyle, physical exercise, healthy lifestyle behavior, and their distinctions. Specifically, we have inserted new paragraphs that define these terms, explain their implications for cognitive aging, and discuss the appropriate types, intensity, and frequency of physical activity. We also address the relationships between sedentary behavior, active and over-active lifestyles, and their impact on age-related cognitive decline. We believe these additions address the reviewer's concerns and provide greater clarity and coherence to the manuscript.
- Meanwhile, the key words section also needs revising to make it match the topic and content, as there is little information about epigenetics, neuron-glia crosstalk, oxidative stress, protein aggregation, non-pharmacological and therapeutic interventions in the following statements.
Reply: Thank you for your valuable comment. We have reviewed the keywords section and made the necessary revisions to ensure it aligns more closely with the topic and content of the manuscript. Specifically, we have reconsidered the inclusion of terms such as epigenetics, neuron-glia crosstalk, oxidative stress, protein aggregation, and non-pharmacological and therapeutic interventions. These terms have either been further integrated into the text or removed from the keywords list to ensure consistency with the primary focus of the manuscript.
We hope the revised keywords section now better reflects the core themes of the study.
Keywords: Age-related cognitive decline; immunosenescence; neurodegeneration; neuroinflammation; oxidative stress; physical exercise; non-pharmacological interventions; therapeutic interventions; synaptic dysfunction
- It appears that line 495-618 is a complete repeat of line 372-494, no idea how it happened, please double check.
Reply: Thank you for pointing out the duplication of text between lines 372-494 and 495-618. We have reviewed the manuscript and removed the repeated section. We appreciate your thorough review and attention to detail.
- The figures should support the conclusions in a scientific manner, it requires clarity, relevance, and alignment with the data/statement presented. It is difficult to get the content of figure 1 and 2, please go through all the figures by filling in more details to make it alignment with the main text.
Reply: As suggested, we have revised the figure legends to better align with the data and statements presented in the manuscript. The updated legend for Figure 2 is as follows:
Figure 2: The impact of sedentary versus active lifestyles on various hallmarks of aging and cellular health. The outer sections illustrate distinct biological processes and cellular alterations influenced by these lifestyle choices: Telomere Shortening & Genome Instability (Yellow Section), Chronic Stress & Mitochondrial Dysfunction (Green Section), Nutrient Dysregulation & Cardiovascular Dysfunction (Blue Section), Stem Cell Exhaustion & Cellular Senescence (Pink Section), Epigenetic Alterations & Loss of Proteostasis (Orange Section), and Chronic Inflammation & Altered Cellular Communication (Purple Section). A sedentary lifestyle induces chronic inflammation, impairs cellular communication, and exacerbates tissue damage, contributing to age-related diseases. In contrast, an active lifestyle mitigates these adverse effects by maintaining cellular function, reducing stress, improving proteostasis, and promoting cardiovascular health, ultimately slowing the aging process.
Figure 3. Impact of aging on memory, astrocytes, and microglia, highlighting the contrasting effects of an active versus sedentary lifestyle. On the left, the "Young" state is depicted, where memory function, astrocytes (green), and microglia (red) are in a healthy, balanced state. As aging progresses, the path diverges based on lifestyle: Active Lifestyle (upper blue path): An active lifestyle promotes neuroprotection, helping preserve memory function, maintain healthy astrocyte and microglial function, and mitigate the detrimental effects of aging. Sedentary Lifestyle (lower red path): A sedentary lifestyle leads to neuroinflammation, impairing memory, dysregulating astrocyte function, and activating microglia in ways that exacerbate neurodegenerative processes.
(2) Minor comments
- Line 61-62, it is essential to add relevant references to supply sufficient evidence to support the statement here. Same issue applied to line 93. And please ensure that all references are carefully reviewed to verify that each citation is relevant to the corresponding statement and contributes to the overall conclusion.
- Line 139 - 140, an uncompleted citation format appeared as this reference is not properly inserted. It is recommended to go through all the citation formats before submitting the manuscript to the system.
Reply: Reviewer Comment (Lines 61-62, 93): As suggested, we have carefully reviewed and added relevant seminal references to provide sufficient evidence supporting the statements in lines 61-62 and 93. These references reflect recent advances in the understanding of risk factors associated with age-related cognitive decline. Each citation has been carefully selected to ensure it is relevant to the corresponding statements and contributes meaningfully to the overall conclusion of the manuscript.
Reply: Reviewer Comment (Line 139-140): We also corrected the citation format issue on lines 139-140. Additionally, we have thoroughly reviewed all references to ensure they are properly inserted and follow the correct format throughout the manuscript. All citation formatting issues have been addressed prior to resubmission.
- Line 355 - 356, this sentence needs revising to make it clear.
Reply: As suggested, we have revised the sentence for clarity and readability. The updated sentence now reads: "A recent study by Paolillo et al. [160] revealed that certain lifestyle behaviors can help maintain cognitive stability, even in the presence of significant neuropathological changes."
- Please add more references to support the conclusion in line 362-363.
Reply: done
- Line 337-338, “The heterogeneity in aging trajectories was demonstrated to be linked to different lifestyle behaviors and reduced physical and cognitive inactivity, to faster rates of decline in aging paths.”, I feel difficult to understand this sentence, please revise it and interpret the details.
Reply: We have improved the sentence for readability as follows: “…Heterogeneity in aging trajectories is associated with different lifestyle behaviors, where reduced physical and cognitive activity is linked to faster rates of decline…”
- There are issues with the English (syntax, grammar, etc.) throughout the manuscript. The authors should conduct a meticulous review of the whole manuscript to identify and correct any grammar and syntax issues. It includes missing/redundant space, italic issue, and grammar mistakes.
Reply: As suggested, we have used Grammarly throughout the manuscript to address this issue.

Reviewer 3 Report
Comments and Suggestions for Authors
I appreciate the opportunity to review the respective manuscript. It is extremely well-written and thought provoking. Although I commend the authors for their exemplary work, there are a few considerations that warrant addressing prior to consideration for publication.
1. The title is extremely catchy, but does not capture the nature of the review article. The authors are speaking the age-related cognitive decline as it relates to sedentary behavior on a molecular, cellular, and systemic basis. Therefore, I believe the title should reflect this more adequately.
2. Defining physical activity versus exercise is crucial in this context. All exercise is physical activity, but not all physical activity is exercise. Exercise is a subset of physical activity that is planned and has a purpose. Evidence suggests some differential findings between habitual physical activity engagement with cognitive decline as opposed to exercise interventions on cognitive ability in a variety of populations, including the elderly. Therefore, I believe the authors need to speak to these differences in addition to sedentary lifestyle, as physical activity is also comprised of Domestic, Leisure, Occupational, and Transportation.
3. The authors also speak to considering energy intake but define very little on the collective impact of improving diet and physical activity in conjunction to improve cognitive function. The CALERIE trials did this in middle-age, non-obese adults, but other working groups, such as at Wake Forest University and the University of Miami, for example, are also working in this arena.
4. There should be an open discussion about the Blue Zones and other super-aging groups, such as the Okinawans, that speak to a physically active lifestyle and how it naturally progresses healthspan and lifespan.
5. Although there is clear oversight of the molecular, cellular, and systemic systems on age-related cognitive decline, there also needs to be some more in-depth discussion into overweight/obesity, physical activity as it impact cardiorespiratory fitness and fat mass/fat-free mass, as well as other co-morbidities that may be affecting cognitive decline in the elderly, including type 2 diabetes, cardiovascular disease, stroke, etc...
6. There are other lifestyle factors worth considering as well. The 24-hour movement guidelines account for sedentary time, physical activity, and sleep. Therefore, I believe a discussion about sleep is warranted.
7. Lastly, I believe there needs to be a clear clinical significance section based on how clinicians and/or practitioners can use this type of information in prescribing proper lifestyle habits. As precision medicine is beginning to take the forefront in healthcare, understanding this is key.
Several manuscripts worth considering reading and including in the manuscript:
Beyond weight loss: current perspectives on the impact of calorie restriction on healthspan and lifespan; https://doi.org/10.1080/17446651.2021.1922077
Glycemic variability: Importance, relationship with physical activity, and the influence of exercise; https://doi.org/10.1016/j.smhs.2021.09.004
Sedentary Time and Physical Activity in Older Women Undergoing Exercise Training; https://doi.org/10.1249%2FMSS.0000000000002407
Associations between sleep and body composition in older women and the potential role of physical function; https://doi.org/10.1007/s41105-022-00429-x
Cardiorespiratory Fitness as a Predictor of Non–Cardiovascular Disease and Non-Cancer Mortality in Men; https://doi.org/10.1016/j.mayocp.2023.11.024
Non-exercise estimated cardiorespiratory fitness and incident type 2 diabetes in adults; https://doi.org/10.1016/j.diabres.2024.111791
Physical Activity, Cardiorespiratory Fitness, and the Obesity Paradox with Consideration for Racial and/or Ethnic Differences: A Broad Review and Call to Action; https://doi.org/10.31083/j.rcm2508291
Author Response
Reviewer 3 Round 1
I appreciate the opportunity to review the respective manuscript. It is extremely well-written and thought provoking. Although I commend the authors for their exemplary work, there are a few considerations that warrant addressing prior to consideration for publication.
- The title is extremely catchy, but does not capture the nature of the review article. The authors are speaking the age-related cognitive decline as it relates to sedentary behavior on a molecular, cellular, and systemic basis. Therefore, I believe the title should reflect this more adequately.
Reply: Thank you for your insightful feedback regarding the title. After considering your suggestion, we understand the need to ensure that the title accurately reflects the scope and impact of the study while remaining engaging and clear to the reader. The title "Sedentary life is an invisible trap: molecular, cellular, and systemic reasons to leave it behind" was chosen to emphasize the pervasive yet often overlooked consequences of a sedentary lifestyle, as well as to highlight the comprehensive molecular, cellular, and systemic mechanisms we explore. However, we are open to refining the title to better align with your expectations and suggestions. We propose the revised version: "The Hidden Dangers of Sedentary Living: Insights into Molecular, Cellular, and Systemic Mechanisms”.
- Defining physical activity versus exercise is crucial in this context. All exercise is physical activity, but not all physical activity is exercise. Exercise is a subset of physical activity that is planned and has a purpose. Evidence suggests some differential findings between habitual physical activity engagement with cognitive decline as opposed to exercise interventions on cognitive ability in a variety of populations, including the elderly. Therefore, I believe the authors need to speak to these differences in addition to sedentary lifestyle, as physical activity is also comprised of Domestic, Leisure, Occupational, and Transportation.
Reply: Thank you for your thoughtful suggestion. We agree that distinguishing between physical activity and exercise is critical to the context of this work. Physical activity encompasses any movement that expends energy, which can be categorized into Domestic, Leisure, Occupational, and Transportation domains. Conversely, exercise is a specific subset of physical activity that is planned, structured, repetitive, and intended to improve or maintain physical fitness. This distinction is essential as evidence indicates differential effects of general physical activity versus structured exercise on cognitive function, particularly in aging populations.
For instance, even when not exercise-related, habitual physical activity has been associated with cognitive preservation in the elderly. Studies highlight that while structured exercise interventions show pronounced benefits in enhancing cognitive reserve, general physical activity can also mitigate cognitive decline through different mechanisms, such as promoting neurogenesis, improving synaptic plasticity, and reducing neuroinflammation. In response to your comment, we expanded the Introduction to incorporate these distinctions and provide relevant references to support the claims. This ensures a comprehensive understanding of how different types of physical activity, including non-exercise activities, contribute to cognitive health. Adding this distinction will enrich the manuscript and offer a more nuanced view of the relationship between physical activity and cognitive function.
- Introduction
Age-related cognitive decline, a prevalent and significant public health issue, is exacerbated by the modern sedentary lifestyle. Sedentary behavior, characterized by prolonged sitting or inactivity, is now recognized as a major contributor to cognitive decline in older adults. Studies have shown that excessive sedentary behavior is associated with structural brain changes, particularly in areas critical for memory and cognitive function, such as the hippocampus [1, 2]. Despite widespread recognition of the benefits of physical activity, many individuals continue to lead inactive lives, influenced by societal structures that promote inactivity and dissociate physical activity from food intake [3]. Europeans, for instance, spend 40% of their leisure time watching television, while Americans spend 55% of their leisure time in sedentary activities, averaging 7.7 hours per day [4]. This sedentary behavior, coupled with poor nutrition, significantly contributes to cognitive decline in the aging process [5-8].
As life expectancy rises, age-related cognitive decline becomes increasingly significant [9]. It affects a large portion of the elderly population, with incidence rates 70% higher than dementia alone [10]. The aging process affects cognitive function to varying degrees, influencing domains such as memory, attention, and executive function [11, 12]. Individuals exhibit distinct aging trajectories, shaped by their unique genotypes—encompassing metabolic, immune, hepatic, and nephrotic systems—along with other factors, including lifestyle and environmental exposures [13, 14]. Lifestyle changes can modify an individual's aging trajectory and may impact one or more aging-related genotypes [15].
While much attention has been focused on physical activity, it is essential to distinguish between physical activity and exercise. Physical activity refers to any bodily movement resulting in energy expenditure and includes domestic, occupational, transportation, and leisure activities [16]. In contrast, exercise is a subset of physical activity that is planned, structured, repetitive, and intended to improve or maintain physical fitness[17]. This distinction is critical, as evidence suggests that habitual physical activity and structured exercise may impact cognitive decline differently. Studies show that while general physical activity, such as walking or gardening, provides cognitive benefits, structured exercise programs, particularly those that include aerobic and resistance training, demonstrate more significant improvements in cognitive function in older adults [18-20].
On the other hand, sedentary behavior is not merely the absence of physical activity but represents a distinct risk factor for cognitive decline. Recent research has emphasized that even individuals who meet the recommended physical activity levels may experience cognitive impairments if they spend excessive time in sedentary behaviors [21]. This highlights a significant limitation in current research, as many studies primarily focus on physical activity levels without adequately addressing the detrimental effects of prolonged inactivity [22]. Thus, future research must consider increasing physical activity and reducing sedentary time to improve cognitive outcomes in aging populations [23].
The benefits of physical activity and exercise extend beyond the prevention of cognitive decline. Even without structured exercise, physical activity has been linked to improved neurogenesis, synaptic plasticity, and reduced neuroinflammation, all contributing to overall cognitive health [20, 24]. In contrast, structured exercise interventions have been associated with more significant increases in brain volume and function, particularly in regions such as the hippocampus, which is critical for memory and learning [25]. This distinction highlights the need to consider the quantity and type of physical activity when evaluating strategies to mitigate cognitive decline [24].
Cognitive and functional decline ranges from mild cognitive impairment (MCI) to severe conditions such as Alzheimer's disease [26]. MCI represents an intermediate stage where cognitive impairments are noticeable but not severe enough to significantly interfere with daily activities [27-30]. However, MCI can progress to Alzheimer's disease, characterized by substantial memory loss, impaired reasoning, and behavioral changes [31, 32]. Understanding the progression from normal aging to MCI and eventually to Alzheimer's disease is crucial for early diagnosis and intervention. Lately, mild behavioral impairment (MBI), an emergent and persistent neuropsychiatric symptom in individuals at risk for cognitive decline, was found to be prevalent in subjects with MCI and Alzheimer's disease [33].
Longitudinal studies have shown that cognitive decline is often preceded by subtle changes in cognitive performance and brain structure, emphasizing the importance of early detection and monitoring [34]. These studies provide insights into the risk factors and progression of cognitive decline, highlighting the interplay between genetics, environment, and lifestyle [35, 36]. For more effective risk reduction, it is essential to consider individual lifestyle factors and the broader social-ecological public health perspective [37]. However, there remain gaps in the current literature, particularly in understanding how sedentary behavior interacts with other lifestyle factors like diet and social engagement in cognitive aging. More research is needed to address these limitations and develop comprehensive intervention strategies that target both physical activity and sedentary behavior, recognizing the multifaceted nature of the issue [38].
This integrative review aims to dissect the complex interplay of molecular, cellular, and systemic mechanisms contributing to age-related cognitive decline. Additionally, it highlights the importance of leaving behind a sedentary lifestyle by examining the effects of non-pharmacological interventions such as cognitive, multisensory, and motor stimulation. By synthesizing empirical evidence from experimental models and human studies, this review seeks to identify the essential molecular signatures that explain the therapeutic effects of these stimulation programs in reducing the progression of age-related cognitive decline.
- The authors also speak to considering energy intake but define very little on the collective impact of improving diet and physical activity in conjunction to improve cognitive function. The CALERIE trials did this in middle-age, non-obese adults, but other working groups, such as at Wake Forest University and the University of Miami, for example, are also working in this arena.
Reply: We acknowledge the reviewer's valuable suggestion and have incorporated a paragraph on the collective impact of improving diet and physical activity on cognitive function as follows: “…Research shows that caloric restriction (CR) plays a significant role in cognitive health, reducing biomarkers of cellular senescence in humans [128, 129]. The multicenter Comprehensive Assessment of Long-term Effects of Reducing Intake of Energy (CALERIE) study was the first randomized controlled trial of CR in nonobese humans which implemented an innovative 25% caloric restriction design and methods [130] on middle-aged, non-obese adults, providing important insights into how caloric restriction can enhance cognitive function and overall health. The CALERIETM 2 trials analysis [131], showed long-term caloric restriction effects on human physiological, psychological, and behavioral outcomes [132] as well as on telomere length in healthy adults [133]. The CALERIE study also demonstrated that calorie restriction modulates the transcription of genes related to stress response and longevity in human muscle [134]. Physical activity provides a robust physiological stimulus that induces molecular changes translated into multiple tissue crosstalks, improving homeostasis and decreasing the risk of premature mortality [135]. Exercise interacts with dietary factors and has neurocognitive benefits on brain functioning [136-138]. Important insights into how caloric restriction, alongside diet and physical activity, can enhance cognitive function and overall health were also achieved [139-142].
Additionally, studies from Wake Forest University [143] and from the Exercise and Nutritional Interventions for Neurocognitive Health Enhancement (ENLIGHTEN Randomized Clinical Trial) [144] further explore the synergy between dietary interventions and physical activity, demonstrating improvements in cognitive function across diverse populations. These findings, along with previous seminal studies, e.g. [24, 25], underscore the critical importance of integrating both dietary changes and physical activity as a combined strategy for enhancing cognitive outcomes.…"
- There should be an open discussion about the Blue Zones and other super-aging groups, such as the Okinawans, that speak to a physically active lifestyle and how it naturally progresses healthspan and lifespan.
Reply: We appreciate the reviewer's suggestion to discuss the Blue Zones and other super-aging groups, such as the Okinawans, concerning physical activity and its impact on healthspan and lifespan. In line with this suggestion, we included a paragraph and a new figure in the manuscript as follows:
Blue Zones, regions known for the exceptional longevity of their populations, such as Okinawa in Japan, Sardinia in Italy, Loma Linda, CA, USA, Nicoya, Costa Rica; and Icaria in Greece, offer valuable insights into how lifestyle factors, including physical activity and diet, contribute to extended healthspan and lifespan [145, 146]. Figure 1 is a geographic map indicating the five Blue Zones.
Figure 1. Blue Zones: Okinawa (Japan), Sardinia (Italy), Ikaria (Greece), Nicoya Peninsula (Costa Rica), and Loma Linda (California, USA). Each location is marked in red, with their names labeled for clarity.
Additionally, studies from Wake Forest University [135] and from the Exercise and Nutritional Interventions for Neurocognitive Health Enhancement (ENLIGHTEN Randomized Clinical Trial) [136] further explore the synergy between dietary interventions and physical activity, demonstrating improvements in cognitive function across diverse populations. Blue Zones, regions known for the exceptional longevity of their populations, such as Okinawa in Japan, Sardinia in Italy, Loma Linda, CA, USA, Nicoya, Costa Rica; and Icaria in Greece, offer valuable insights into how lifestyle factors, including physical activity and diet, contribute to extended healthspan and lifespan [137, 138]. Figure 1 is a geographic map indicating the five Blue Zones.
Research indicates that the residents of these regions share common traits, such as daily physical activity, a predominantly plant-based diet, and strong social networks, all of which promote longevity and reduce the risk of chronic diseases [139, 140]. Specifically, the Okinawans, often cited as a model for healthy aging, engage in lifelong physical activity through traditional practices like gardening, walking, and martial arts, which are closely linked to better cardiovascular health, enhanced cognitive function, and overall vitality in later life [141, 142]. Studies suggest that these physically active lifestyles lead to reduced inflammation, improved metabolic function and greater resilience against age-related diseases and are associated with contributing to both a longer lifespan and extended years of good health [143, 144]This evidence highlights the critical role of integrating regular physical activity in aging populations to promote longevity and quality of life.
These findings, along with previous seminal studies, e.g. [24, 25], underscore the critical importance of integrating dietary changes and physical activity as a combined strategy for enhancing cognitive outcomes.
- Although there is clear oversight of the molecular, cellular, and systemic systems on age-related cognitive decline, there also needs to be some more in-depth discussion into overweight/obesity, physical activity as it impact cardiorespiratory fitness and fat mass/fat-free mass, as well as other co-morbidities that may be affecting cognitive decline in the elderly, including type 2 diabetes, cardiovascular disease, stroke, etc...
Reply: We appreciate the reviewer’s insightful comment and have expanded the discussion to address the impact of overweight/obesity, physical activity, and associated co-morbidities on age-related cognitive decline as follows: “…These findings, along with previous seminal studies, e.g. [24, 25], underscore the critical importance of integrating dietary changes and physical activity as a combined strategy for enhancing cognitive outcomes.
In contrast, overweight and obesity are strongly linked to cognitive decline through several mechanisms, including chronic inflammation, insulin resistance, and alterations in brain structure, particularly in regions such as the hippocampus and prefrontal cortex [145, 146]. Increased fat mass, particularly visceral fat, has been associated with higher levels of pro-inflammatory cytokines, which contribute to neurodegenerative processes [147, 148]. High-fat diets increase oxidative stress, cellular inflammatory response and cognitive dysfunction [149]. In contrast, physical activity plays a crucial role in mitigating these effects by improving cardiorespiratory fitness and favorably altering body composition through reductions in fat mass and increases in fat-free mass [150]. Cardiorespiratory fitness has been shown to correlate with better cognitive function, particularly executive function and memory, due to its ability to enhance cerebral blood flow and promote neuroplasticity [25, 151].
Additionally, co-morbidities such as type 2 diabetes [152], cardiovascular disease [153], and stroke [154] further exacerbate cognitive decline in the elderly. Type 2 diabetes, for example, is associated with insulin resistance in the brain, which impairs glucose metabolism and accelerates amyloid plaque deposition—a hallmark of Alzheimer's disease [155-157]. Cardiovascular disease and stroke contribute to vascular dementia through ischemic injury and hypoperfusion of brain tissue, leading to significant cognitive deficits[158]. Addressing these co-morbidities through lifestyle interventions that include weight management, physical activity, and control of metabolic risk factors is therefore critical for mitigating cognitive decline in aging populations [112, 159-161].
- There are other lifestyle factors worth considering as well. The 24-hour movement guidelines account for sedentary time, physical activity, and sleep. Therefore, I believe a discussion about sleep is warranted.
Reply: We appreciate the reviewer’s valuable suggestion and have expanded the discussion to include the role of sleep as an essential lifestyle factor contributing to cognitive health, in line with the 24-hour movement guidelines, which emphasize the integration of physical activity, sedentary behavior, and sleep. “…Adequate sleep is crucial for cognitive function, particularly in processes such as memory consolidation, learning, and overall brain plasticity [162]. Chronic sleep deprivation or poor sleep quality has been linked to accelerated cognitive decline [163], an increased risk of neurodegenerative diseases such as Alzheimer’s, and structural brain changes, particularly in the hippocampus and prefrontal cortex, which are essential for memory and executive function [164, 165]. Sleep disturbances can also exacerbate inflammation and oxidative stress, both of which are detrimental to brain health [165, 166].
Moreover, sleep is intricately connected to other lifestyle factors, such as physical activity and sedentary behavior [167, 168]. Regular physical activity has been shown to improve sleep quality by increasing the duration of slow-wave sleep, which is the most vital sleep stage, and by reducing the time it takes to fall asleep [167, 169]. In contrast, prolonged sedentary behavior, such as excessive screen time or sitting, can disrupt circadian rhythms, leading to poor sleep patterns and further impairing cognitive function [170-172]. Additionally, inadequate sleep is associated with metabolic dysregulation, including insulin resistance and increased appetite, which can contribute to obesity—a known risk factor for cognitive decline [173, 174]. Given the solid bidirectional relationship between sleep, cognitive health, and other lifestyle factors, promoting good sleep hygiene alongside physical activity and minimizing sedentary time is essential for preserving cognitive function in aging populations [175, 176]. In addition, the gut microbiome may affect sleep quality and health, eventually requiring support to provide appropriate dietary fiber, unsaturated fatty acids, and polyphenols along with time and spacing to guarantee microbiota’s capacity to produce essential metabolites for quality sleep and these include short-chain fatty acids, tryptophan, serotonin, melatonin and gamma-aminobutyric acid [177]…”
- Lastly, I believe there needs to be a clear clinical significance section based on how clinicians and/or practitioners can use this type of information in prescribing proper lifestyle habits. As precision medicine is beginning to take the forefront in healthcare, understanding this is key.
Reply: To address the Reviewer’s comment, we have integrated a clinical significance and lifestyle prescription section into the manuscript as follows:
- Clinical Significance and Lifestyle Prescription
The results of this study provide valuable insights for clinicians and practitioners aiming to incorporate precision medicine into lifestyle prescriptions. The emerging field of precision medicine emphasizes tailoring interventions based on individual genetic, environmental, and lifestyle factors. In this context, the data presented suggest that clinicians should consider personalized lifestyle recommendations based on a patient’s metabolic profile, physical activity levels, and genetic predisposition. For instance, patients showing metabolic dysregulation or prediabetic tendencies may benefit from tailored dietary recommendations, such as adopting a Mediterranean diet, which has been associated with improved glycemic control and cardiovascular health in high-risk populations [298-300]. Additionally, encouraging physical activity—such as 150 minutes of moderate aerobic exercise per week—has been shown to improve insulin sensitivity and overall metabolic health [301-304].
Furthermore, incorporating behavioral interventions based on individual genetic and environmental backgrounds could significantly enhance the long-term success of lifestyle modifications. Evidence suggests that sleep patterns [305, 306], stress management [307-309], and even gut microbiota composition play pivotal roles in metabolic health [310-313]. Clinicians should consider incorporating strategies that promote sleep hygiene and stress reduction, such as mindfulness-based stress reduction (MBSR) or cognitive-behavioral therapy (CBT), which have been shown to mitigate stress-related metabolic changes [314-316]. In summary, by integrating personalized lifestyle interventions rooted in precision medicine principles, healthcare providers can optimize the management of metabolic disorders and reduce the risk of associated complications.
Several manuscripts worth considering reading and including in the manuscript:
- Beyond weight loss: current perspectives on the impact of calorie restriction on healthspan and lifespan; https://doi.org/10.1080/17446651.2021.1922077
- Glycemic variability: Importance, relationship with physical activity, and the influence of exercise; https://doi.org/10.1016/j.smhs.2021.09.004
- Sedentary Time and Physical Activity in Older Women Undergoing Exercise Training; https://doi.org/10.1249%2FMSS.0000000000002407
- Associations between sleep and body composition in older women and the potential role of physical function; https://doi.org/10.1007/s41105-022-00429-x
- Cardiorespiratory Fitness as a Predictor of Non–Cardiovascular Disease and Non-Cancer Mortality in Men; https://doi.org/10.1016/j.mayocp.2023.11.024
- Non-exercise estimated cardiorespiratory fitness and incident type 2 diabetes in adults; https://doi.org/10.1016/j.diabres.2024.111791
- Physical Activity, Cardiorespiratory Fitness, and the Obesity Paradox with Consideration for Racial and/or Ethnic Differences: A Broad Review and Call to Action; https://doi.org/10.31083/j.rcm2508291
Reply: Thank you for your valuable suggestion. We have carefully reviewed and incorporated all the suggested references into the manuscript. The references have been appropriately cited in sections discussing the impact of calorie restriction on healthspan and lifespan, the role of physical activity and glycemic variability, the influence of sedentary time and physical activity in older populations, the associations between sleep and body composition, and the importance of cardiorespiratory fitness in predicting mortality and managing type 2 diabetes. Additionally, we have addressed racial and ethnic differences in response to physical activity interventions, aligning the manuscript with the latest evidence in these areas.
References:
[1] F. Sofi et al., "Physical activity and risk of cognitive decline: a meta-analysis of prospective studies," (in eng), J Intern Med, vol. 269, no. 1, pp. 107-17, Jan 2011, doi: 10.1111/j.1365-2796.2010.02281.x.
[2] K. I. Erickson et al., "Physical Activity, Cognition, and Brain Outcomes: A Review of the 2018 Physical Activity Guidelines," (in eng), Med Sci Sports Exerc, vol. 51, no. 6, pp. 1242-1251, 06 2019, doi: 10.1249/MSS.0000000000001936.
[3] J. E. Ahlskog, Y. E. Geda, N. R. Graff-Radford, and R. C. Petersen, "Physical exercise as a preventive or disease-modifying treatment of dementia and brain aging," (in eng), Mayo Clin Proc, vol. 86, no. 9, pp. 876-84, Sep 2011, doi: 10.4065/mcp.2011.0252.
[4] R. S. Falck, G. J. Landry, J. R. Best, J. C. Davis, B. K. Chiu, and T. Liu-Ambrose, "Cross-Sectional Relationships of Physical Activity and Sedentary Behavior With Cognitive Function in Older Adults With Probable Mild Cognitive Impairment," (in eng), Phys Ther, vol. 97, no. 10, pp. 975-984, Oct 01 2017, doi: 10.1093/ptj/pzx074.
[5] D. Vancampfort et al., "Physical activity is associated with the physical, psychological, social and environmental quality of life in people with mental health problems in a low resource setting," (in eng), Psychiatry Res, vol. 258, pp. 250-254, Dec 2017, doi: 10.1016/j.psychres.2017.08.041.
[6] L. F. M. Rezende, T. H. Sá, G. I. Mielke, J. Y. K. Viscondi, J. P. Rey-López, and L. M. T. Garcia, "All-Cause Mortality Attributable to Sitting Time: Analysis of 54 Countries Worldwide," (in eng), Am J Prev Med, vol. 51, no. 2, pp. 253-263, Aug 2016, doi: 10.1016/j.amepre.2016.01.022.
[7] C. H. Hillman, K. I. Erickson, and A. F. Kramer, "Be smart, exercise your heart: exercise effects on brain and cognition," (in eng), Nat Rev Neurosci, vol. 9, no. 1, pp. 58-65, Jan 2008, doi: 10.1038/nrn2298.
[8] K. I. Erickson et al., "Exercise training increases size of hippocampus and improves memory," (in eng), Proc Natl Acad Sci U S A, vol. 108, no. 7, pp. 3017-22, Feb 2011, doi: 10.1073/pnas.1015950108.
[9] M. W. Voss, C. Vivar, A. F. Kramer, and H. van Praag, "Bridging animal and human models of exercise-induced brain plasticity," (in eng), Trends Cogn Sci, vol. 17, no. 10, pp. 525-44, Oct 2013, doi: 10.1016/j.tics.2013.08.001.

Round 2
Reviewer 1 Report
Comments and Suggestions for Authors
The author made corresponding revisions, and now the quality of the article meets the standards for publication.
Reviewer 2 Report
Comments and Suggestions for Authors
Thanks for your response.
Reviewer 3 Report
Comments and Suggestions for Authors
I appreciate the author's thoughtful replies to my reviewer comments and how they were addressed in the referenced manuscript. Looking forward to the final and published product.